# Developmental and Molecular Effects of C-Type Natriuretic Peptide Supplementation in In Vitro Culture of Bovine Embryos

**DOI:** 10.3390/ijms252010938

**Published:** 2024-10-11

**Authors:** Camila Bortoliero Costa, Nathália Covre da Silva, Amanda Nespolo Silva, Elisa Mariano Pioltine, Thaisy Tino Dellaqua, Amanda Fonseca Zangirolamo, Flávio Vieira Meirelles, Marcelo Marcondes Seneda, Marcelo Fábio Gouveia Nogueira

**Affiliations:** 1Graduate Program in Pharmacology and Biotechnology, Institute of Biosciences, São Paulo State University (UNESP), Botucatu 18618-970, SP, Brazil; thaisydellaqua@hotmail.com; 2Department of Biological Sciences, School of Sciences, Humanities and Languages, São Paulo State University (UNESP), Assis 19806-900, SP, Brazil; 3Laboratory of Animal Reproduction, Department of Veterinary Medicine, University of Londrina (UEL), Londrina 86057-970, PR, Brazil; 4Graduate Program in Anatomy of Domestic and Wild Animals, University of São Paulo (USP), Pirassununga 13635-000, SP, Brazil; 5Department of Veterinary Medicine, College of Animal Science and Food Engineering, University of São Paulo (USP), Pirassununga 13635-000, SP, Brazil; 6National Institute of Science and Technology for Dairy Production Chain (INCT–LEITE), University of Londrina (UEL), Londrina 86057-970, PR, Brazil

**Keywords:** C-type natriuretic peptide, embryonic metabolism, NPR2, transcript abundance, cattle

## Abstract

The use of C-type natriuretic peptide (CNP) in the interaction with the oocyte and in the temporary postponement of spontaneous meiosis resumption has already been well described. However, its action in pre-implantation developmental-stage embryos is yet to be understood. Thus, our study aimed to detect the presence of the canonical CNP receptor (natriuretic peptide receptor, NPR2) in germinal vesicle (GV)-, metaphase II (MII)-, presumptive zygote (PZ)-, morula (MO)-, and blastocyst (BL)-stage embryos and, later, to observe possible modulations on the embryos when co-cultured with CNP. In Experiment I, we detected and quantified NPR2 on the abovementioned embryo stages. Further, in Experiment II, we intended to test different concentrations (100, 200, or 400 nM of CNP) at different times of inclusion in the in vitro culture (IVC; inclusion from the beginning, i.e., day 1, or from day 5). In Experiment III, 400 nM of CNP was used on day 1 (D1) in the IVC, which was not demonstrated to be embryotoxic, and it showed potentially promising results in the blastocyst production rate when compared to the control. Thus, we analyzed the embryonic development rates of bovine embryos (D7) and hatching kinetics (D7, D8, and D9). Subsequently, morula and blastocyst were collected and evaluated for transcript abundance of their competence and quality (apoptosis, oxidative stress, proliferation, and differentiation) and lipid metabolism. Differences with probabilities less than *p* < 0.05, and/or fold change (FC) > 1.5, were considered significant. We demonstrate the presence of NPR2 until the blastocyst development stage, when there was a significant decrease in membrane receptors. There was no statistical difference in the production rate after co-culture with 400 nM CNP. However, when we evaluated the abundance of morula transcripts, there was an upregulated transcription in *ADCY6* (*p* = 0.057) and downregulated transcripts in *BMP15* (*p* = 0.013), *ACAT1* (*p* = 0.040), and *CASP3* (*p* = 0.082). In addition, there was a total of 12 transcriptions in morula that presented variation FC > 1.5. In blastocysts, the treatment with CNP induced upregulation in *BID*, *CASP3*, *SOX2*, and *HSPA5* transcripts and downregulation in *BDNF*, *NLRP5*, *ELOVL1*, *ELOVL4*, *IGFBP4*, and *FDX1* transcripts (FC > 1.5). Thus, our study identified and quantified the presence of NPR2 in bovine pre-implantation embryos. Furthermore, 400 nM of CNP in IVC, a concentration not previously described in the literature, modulated some transcripts related to embryonic metabolism, and this was not embryotoxic morphologically.

## 1. Introduction

The C-type natriuretic peptide (CNP) molecule plays a central role in regulating the meiotic progress of the oocyte into growing follicles in mammals [1,2,3,4,5,6]. However, the relationship between CNP and the embryo has few reports in the literature [5,6,7].

In cattle, it has been detected that CNP is physiologically produced by granulosa and *cumulus* cells and by oocytes [8] and in extracellular vesicles [9]. In addition to the identification of endogenous CNP, when included, CNP in in vitro maturation (IVM) can affect lipid metabolism in the oocyte and embryo [5], and its inclusion in the in vitro culture (IVC) can alter some target genes and the lipid profile [7].

In contrast, CNP is produced only by granulosa cells in mice, and it binds to a preferential receptor CNP type 2 (NPR2) present in *cumulus* cells, oocytes, and pre-implantation embryos [6]. The presence of NPR2 in bovine oocytes has been reported [8], but there are no reports of its presence in embryonic development post fertilization.

There was a report in 2020 [7] where the authors reported that CNP supplementation in an in vitro culture (Day 5) of bovine embryos modulated the lipid profile, as well as the abundance of some transcripts related to embryonic lipid metabolism (*ELOVL6*, *CPT2*, and *SREBP1*). However, there are no studies about possible receptors related to the action of CNP on the bovine embryo, as already described in oocytes. Thus, we aimed to confirm and quantify the NPR2 receptor in oocytes and pre-implantation-stage embryos (Experiment I) to test different concentrations (100, 200, or 400 nM of CNP) at different moments of inclusion in the IVC (Experiment II) and, lastly, to evaluate the action of CNP on blastocyst production and hatching rates, as well as the abundance of transcripts from morula and blastocyst in terms of competence, quality, and lipid metabolism (Experiment III). Therefore, our general objective was to evaluate the effects on the in vitro production of bovine blastocysts using different concentrations of exogenous CNP.

## 2. Results

### 2.1. Immunolocalization of the CNP Receptor

Based on fluorescence, NPR2 could be localized in different stages of embryonic development. NPR2 is located primarily on oocyte membranes at the germinal vesicle stage, germinal vesicle breakdown, and presumptive zygote. Since the receptor is highly expressed in GV oocytes, we used it as a negative control for the immunostaining reaction, omitting the primary antibody. However, the morula stage showed the highest labeling, and the blastocyst stage showed the lowest (Figure 1 and Figure 2; *p* ≤ 0.05).

### 2.2. Dose–Response Effect of CNP on Embryo Production

A moderate statistical significance was observed in the groups treated with CNP (100, 200, and 400 nM) compared to the control group on day 1 (*p* = 0.082), and no significant difference was observed on day 5 (*p* = 0.743) related to the moment of CNP addition on the IVC. The evaluated embryos did not show signs of toxicity (there was no reduction in blastocyst rates and morphological features), so the highest concentration of CNP (400 nM) was chosen to be used together with the control group. Likewise, the use of CNP from the beginning of the culture (day 1) proved to be useful to maintain a regular blastocyst rate (Table 1 and Table 2). Therefore, we chose to keep the embryos exposed to CNP from the beginning of the IVC (day 1) considering the development and activation of the embryonic genome in this period.

### 2.3. Impact of Co-Culture with CNP on Blastocyst Formation and Target Transcripts

A total of 2005 presumptive zygotes were cultured, obtaining a blastocyst rate (D7) of 32.6% and 34.1% in the C-400 (400 nM of CNP) and control groups, respectively (*p* = 0.52). There was no significant difference in the observed hatching rate between groups (*p* = 0.66; Table 3).

Transcript abundance in morula was downregulated in *ACAT1* (*p* = 0.040), *CASP3* (*p* = 0.082), and *BMP15* (*p* = 0.013) and was upregulated in *ADCY6* (*p* = 0.057) (Figure 3). Fold change analysis evidenced a variation in 12 targets (FCs > 1.5; Table 4; Appendix A). In the heatmap, there was discriminant clustering between samples from both groups (Figure 4A). In the Principal Component Analysis (PCA) and Partial Least Squares–Discriminant Analysis (PLS-DA), it was possible to observe an overlap in the PCA plot, and in PLS-DA, there was a separation between the control and CNP-treated group (Figure 4B and Appendix A).

The transcript abundance of blastocyst was not significantly altered. However, when fold change analysis was considered, there were transcripts upregulated in the CNP group, such as *HSPA5*, *SOX2*, *CASP3*, and, *BID*, and downregulated as *BDNF*, *NLRP5*, *AGPAT9*, *IGFBP4*, *ELOVL4*, *ELOVL1*, and *FDX1* (Table 5; Appendix A). In addition, it was possible to observe a modest discrimination of the groups when 2D PLS-DA was applied (Figure 5B) and in a cluster analysis heatmap (Figure 5A).

## 3. Discussion

To our knowledge, this study is the first to detect and quantify the NPR2 (CNP receptor) in bovine pre-implantation embryos. Our results supported the hypothesis that the use of CNP in the culture of bovine embryos would alter the embryonic metabolism, sustained by data from our group that observed changed transcript abundance in embryos submitted to 100 nM of CNP from day 5 of the IVC [7].

In addition, the use of 400 nM of CNP in IVC, although no strong statistical difference in the production rate was observed, showed more promising results in blastocyst production when compared to the control (46.1 ± 7.8 vs. 32.8 ± 14.2, respectively; Table 1). Moreover, changes in the abundance of some transcripts related to lipid metabolism, embryonic development, and oxidative stress were observed in morulas and blastocysts treated with CNP, reinforcing that that molecule possess an active effect on the in vitro culture.

NPR2, when activated by CNP binding, triggers a guanylyl cyclase domain, which generates cGMP. This process causes the elevation of cGMP and transfer through gap junctions from *cumulus* cells to oocytes and induces an inhibitory action on phosphodiesterase 3A (PDE3A), maintaining high cAMP concentrations in oocytes and blocking meiosis resumption [10]. In our study, the presence of NPR2 was observed in all stages of embryonic development analyzed. The detection of NPR2 in oocytes (GV and MII stages) and presumptive zygotes was similarly expressed and quantified. In blastocysts, there was a pronounced decrease in the number of receptors compared to the morula, as equally described by Xi et al. [6] in mouse embryos. In cattle, the presence of NPR2 was observed in oocytes at the germinal vesicle stage in the membrane and, after the resumption of meiosis, there was a decrease in receptor detection in matured oocytes in metaphase II [8]. Thus, our results corroborate those findings by Xi et al. [6] in mice.

Exogenous CNP utilization in pre-IVM and IVM has been a routine for years, with different concentrations and in several species: 100 nM in cattle [3,5,11] 100 nM, 500 nM in murine [6,12], 100 nM in cats [13], and 150 ng/mL in goats [14]. However, the relationship between CNP and the embryo has few reports in the literature [5,6,7]. Most importantly, when looking for its use in the in vitro culture (IVC) step, there is only one study with bovine species [7].

In this context, we supported the hypothesis that the use of CNP in the culture of bovine embryos would alter embryonic metabolism. It was possible to observe changes in the abundance of some transcripts related to lipid metabolism, embryonic development, and oxidative processes stress in morulas and blastocysts treated with CNP. These results support and corroborate the study previously published by our group, which observed changes in the abundance of transcripts in embryos subjected to 100 nM CNP from day 5 of IVC [7]. Moreover, in morulas, the adenylyl cyclase type 6—*ADCY6* (*p* = 0.057) gene was upregulated, and the Bone morphogenic protein 15 *BMP15* (*p* = 0.013), Acetyl-CoA acetyltransferase 1—*ACAT1* (*p* = 0.040), and caspase 3—*CASP3* (*p* = 0.082) genes were downregulated. Further, a total of 12 transcriptions in morula presented a variation if considered an FC > 1.5.

The *ADCY6* gene encodes a member of the adenylyl cyclase family of proteins that are required for the synthesis of cyclic AMP [15]. Adenylyl cyclase is reported to present in 13- to 16-day-old bovine embryos and has been reported to modulate cAMP and cGMP concentrations, which determine the rapid proliferation of embryonic cells or even signaling to the endometrium [16]. In the present study, detection of NPR2 with its potential activity (derived of the modulation observed when the ligand—i.e., CNP—was added to the IVC medium) in a cellular compartment other than the *cumulus* cells assumes a natural function for the NPR2/CNP complex also in pre-implantation embryos, at least in bovine and murine species. It was inferred that the elevation of *ADCY6* transcript abundance occurred due to the exposure of morula to CNP, ultimately triggering greater conversion of ATP to cAMP.

In mammals, *BMP15* is related to oocyte maturation and also to cholesterol biosynthesis, improving oocyte competence and embryonic development in cattle [17,18]. The *BMP15* transcript was relatively elevated in morula that did not receive CNP treatment. Several studies have reported an increase in the *BMP15* transcript during maturation in oocytes from buffaloes, dogs, and cows [18,19,20]. Furthermore, the metabolic pathways of *cumulus* cells, particularly glycolysis and cholesterol biosynthesis, are highly affected when there is a mutation in the *BMP15* gene [21]. Thus, we infer that CNP could be reducing *BMP15* expression by modulating cholesterol biosynthesis through cGMP elevations. This hypothesis is because the CNP treatment did not decrease blastocyst production and hatching rates and did not increase transcripts negatively related to embryonic quality. Also, oocyte maturation, where the role of *BMP15* is more associated, has not been tested, and therefore the role of *BMP15* in the IVC of embryos is an important point highlighted by this study.

A reduction in the abundance of the *ACAT1* transcript was observed in CNP-treated morula. *ACAT1* promote free cholesterol to be esterified into cholesterol esters [22]. In our study, however, we inferred that the reduction in the ACAT1 transcript may have been caused by CNP, which modulated the metabolism of some lipid classes, reducing some cholesterol esters.

Another important point was that CNP-treated morula tended to lower concentrations of *CASP3* transcripts. This gene is directly linked with the cell death program, that is, the apoptosis [23]. *CASP3*, in oocytes, is associated with low competence and death of the oocyte [24,25]. In this context, Kaihola et al. [26] showed that in secretomes from high-quality blastocysts, the levels of *CASP3* were significantly lower than in embryos that became arrested and low-quality blastocysts.

Several studies have demonstrated the interference of CNP in the apoptosis process. In a study with porcine COCs, DNA damage was dramatically decreased due to CNP exposure at different concentrations and for 24 h. Also, CNP exposure significantly downregulated pro-apoptotic genes, i.e., *BAX*, *CASP3*, *C-MYC*, and *P53* [27]. Moreover, on average, *cumulus* cell layers of human COCs cultured for 24 h in the prematuration culture (PMC) supplemented with CNP showed a very low degree of caspase-3/7 activation [28]. In addition, Zhang et al. [29] observed a decrease in the proportion of DNA-fragmented nuclei in blastocysts from sheep oocytes pre-treated with 200 nM CNP for 4 h followed by 24 h IVM, compared with blastocysts from conventional 24, 26, or 28 h IVM.

Furthermore, initial studies suggest that mitochondria play a central role in apoptosis [30], and after their injury, there is a loss of mitochondrial membrane potential and release of factors such as the apoptosis initiation factor and cytochrome C. The release of cytochrome C activates caspase-9, which then activates effector caspases such as *CASP3* [23]. Thus, we inferred that the reduction in the abundance of *CASP3* transcript, observed in the CNP-treated group, potentially made it possible for the morula to have fewer damages associated with cell death.

In addition, in blastocysts treated with CNP, it was possible to observe a reduction, when evaluating the fold change, of targets related to predicting embryo quality, lipid metabolism, fatty acid biosynthesis, elongation, endoplasmic reticulum action, synthesis steroid hormones, and catalyzing cholesterol cleavage (*BDNF*, *NLRP5*, *AGPAT9*, *ELOVL1*, *ELOVL4*, *IGFBP4*, and *FDX1* transcripts). An indication of the decrease in the abundance of the *AGPAT9* transcript was observed in blastocysts treated with CNP. *AGPAT9* has been identified as a key regulator of lipid accumulation in adipocytes [31], suggesting that this may be a biomarker for lipid droplet content in the embryo. In theory, with the reduction in the abundance of *AGPAT9* and other targets, blastocysts cultured with CNP should have a lower lipid content; however, based on the results presented, we were unable to measure its content or lipid profile. Thus, further studies are needed regarding embryonic metabolism in order to understand whether the real changes are caused by the inclusion of CNP.

The family of elongases of very-long-chain fatty acids (*ELOVL*) are enzymes responsible for the condensation reaction necessary for the biosynthesis of long-chain fatty acids (FA). Increased *ELOVL1* expression is directly involved in the elongation of saturated and monounsaturated FAs [32]. The *ELOVL4* is an elongase responsible for the biosynthesis of very-long-chain (VLC, ≥C28) saturated (VLC-SFAs) and polyunsaturated (VLC-PUFAs) fatty acids [33]. It is known that FAs are mainly stored as triacyl glycerides (TAGs, main lipid class found in the cytoplasm of mammalian cells) and enclosed in lipid droplets [34]. Also, their presence may be a compromising factor in cryopreservation processes, increasing risks of cellular injuries [7]. In our results, we showed that *ELOVL4* was upregulated—considering the fold change—therefore, CNP treatment may have potentiated the biosynthesis of long-chain acids. However, we did not observe a reduction in the embryo production rate of CNP-treated embryos. Concomitantly, it was observed that control blastocysts (without CNP treatment) had a greater abundance of pro-apoptotic and apoptosis-related transcripts (*BID* and *CASP3*) than those that received CNP. This fact may suggest a potential protective effect of CNP on cell metabolism in bovine embryos during IVC.

In summary, this study detected and quantified NPR2 (CNP receptor) for the first time, at least to our knowledge, in the pre-implantation stages of the in vitro-produced bovine embryo. Furthermore, it was determined that the use of CNP at a higher concentration than that described in the literature may alter embryonic metabolism based on the abundance of some target transcripts. Also, that CNP concentration did not induce harm when the embryo production rate was observed. Finally, in vivo functional tests will be able to better investigate whether those alterations found in this study could be translated into a better reproductive performance by embryos treated with CNP in the IVC.

## 4. Material and Methods

All animal procedures were approved by the Ethics and Animal Handling Committee of the São Paulo State University (UNESP), Botucatu, São Paulo, Brazil, certificate #1180. The experimental design performed in our study is illustrated below (Figure 6).

### 4.1. In Vitro Production of Embryos

Ovaries from a commercial slaughterhouse, from bovine females with a predominantly *Bos taurus indicus* phenotype of the Nellore breed, were collected, packaged, and transported to the laboratory in 0.9% saline solution between 30 and 35 °C. Follicles with a diameter of 2–8 mm were aspirated with hypodermic needles (30 × 8; 21G) attached to 10 mL syringes for the recovery of *cumulus–oocyte* complexes (COCs), with a maximum interval—between the arrival of the ovaries from the slaughterhouse and the end of aspiration—of four hours. Only COCs of qualities I and II were used, and the classification was performed conforming to the method used in previous studies [24,35].

#### 4.1.1. In Vitro Maturation

Previously to in vitro maturation (IVM), COCs were washed three times in TCM-HEPES 199 supplemented with 10% (*v*/*v*) fetal bovine serum, (FBS), 0.20 nM sodium pyruvate, and 83.4 mg/mL of gentamicin (ABS Global Brasil^®^, Mogi Mirim, São Paulo, Brazil). The COCs were matured in drops of 100 mL of TCM-199 medium bicarbonate supplemented with 10% (*v*/*v*) fetal bovine serum (FBS) and 50 μg of gentamycin/mL (ABS Global Brasil^®^, Mogi Mirim, São Paulo, Brazil) and incubated for 24 h in an environment with maximum humidity, 38.5 °C and 20% O_2_.

#### 4.1.2. In Vitro Fertilization

Finishing the previous step, the COCs were washed in HEPES-buffered TCM-199 medium and transferred to 100 mL droplets of the fertilization medium that consisted of Tris-buffered medium (TBM) supplemented with 8 mg/mL fatty acid-free bovine serum albumin (BSA) and 1 mM glutamine (ABS Global Brasil^®^, Mogi Mirim, São Paulo, Brazil). For fertilization in drops of 100 µL, semen from a single Nellore bull was used (Adamo Fiv Kubera; Code 011NE03127, register ACF 3522, Alta Genetics, Watertown, WI, USA), which was previously validated. The cryopreserved semen was heated at 36 °C for 30 s. Sperm selection was performed by Percoll gradient (ABS Global Brasil^®^; Percoll 45% in the upper part and 90% in the lower) by centrifugation (12,100× *g*, for 5 min), the supernatant (600 µL) was discarded, and the sperm pellet was resuspended in 300 µL of fertilization medium and homogenized. The semen was centrifuged again (8127× *g*, for 2 min) and, after discarding the supernatant, the sperm concentration was adjusted to obtain a final concentration of 1 × 10^6^ motile spermatozoa in each drop containing 20 COCs. They were co-incubated for 20 to 22 h in an environment with maximum humidity, 20% O_2,_ and 38.5 °C. The day of insemination was considered day zero (D0).

#### 4.1.3. In Vitro Culture

For in vitro culture, presumptive zygotes were subjected to the removal of *cumulus* cells by successive pipetting and then incubated in SOF (Synthetic Oviduct Fluid) medium supplemented with 8 mg/mL fatty acid-free BSA (ABS Global Brasil^®^, Mogi Mirim, São Paulo, Brazil) under mineral oil under the same temperature and the gaseous atmospheric condition used in the previous steps. On the first day of culture (D1) or four days later (D5), the structures were divided into experimental groups: control (without the addition of CNP) and CNP groups (C-type natriuretic peptide, Sigma–Aldrich/St. Louis, MO, USA). On D3, 50% of the culture media volume was replaced by fresh media (1st feeding), and the same occurred on D5 (50% of the culture media volume was replaced with fresh SOF medium supplemented with glucose; 2nd feeding). During culture, blastocyst (D7) and hatching (D7, D8, and D9) rates were evaluated.

### 4.2. Experiment I—Detection and Quantification of CNP Receptor

The presence of the CNP receptor (NPR2) in germinal vesicle (GV)-, metaphase II (MII)-stage oocytes, presumptive zygote (PZ)-, morula (MO)-, and blastocyst (BL)-stage was evaluated by immunocytochemistry. For this, the collected oocytes and embryos (*n* = 10 cell/stage) were washed three times with phosphate-buffered saline (PBS; Sigma-Aldrich, St. Louis, MO, USA) and fixed immediately in 4% paraformaldehyde for 20 min. After fixation, the oocytes and embryos were incubated with the primary antibody against NPR2 (HPA011977 Sigma-Aldrich; [6]) diluted 1:100 overnight at 4 °C. Subsequently, oocytes and embryos were washed with PBS, incubated with anti-rabbit secondary antibody (Invitrogen, Carlsbad, CA, USA) diluted 1:200 for 1 h at 37 °C, and nuclei were stained with 4′,6-diamidino-2-phenylindole (DAPI; Sigma-Aldrich) for 5 min. The fluorescent signals were examined using a Leica fluorescence optical microscope (Leica—THUNDER, Leica Microsystems, Wetzlar, Germany) and analyzed by densitometric analysis using ImageJ (Version 1.53). The intensity of DAPI and NPR2 is presented as the mean of the fluorescence intensity in arbitrary units.

### 4.3. Experiment II—Dose Response with Different CNP Concentrations and CNP Activity Moment Effects on the Embryo Production Rate

Initially, two experiments were carried out to test some concentrations (control group and groups with 100, 200, and 400 nM of CNP) in terms of embryotoxicity and to test the inclusion of CNP in the IVC at the beginning of culture (D1) or on D5 of IVC. The blastocyst rate of the treated groups was compared to the control, and a statistical decrease in this rate would be indicative of morphological embryotoxicity, whereas the complete blockage of blastocyst production should be considered a lethal CNP concentration [36,37].

### 4.4. Experiment III—Blastocyst Rate and Abundance of Target Transcripts in Embryos Produced under CNP Co-Culture

After obtaining data from Experiment II, the highest concentration used (400 nM) was not morphologically embryotoxic. Therefore, we used this concentration of CNP from the first day of the IVC (D1) and evaluated the impact on blastocyst production and hatching rate. Furthermore, we evaluated the abundance of transcripts concerning competence and quality (apoptosis, oxidative stress, proliferation, and differentiation) and lipid metabolism in morula (D5) and hatched blastocyst (D7 and D8) from the control group (without CNP) and the group treated with 400 nM CNP.

### 4.5. Reverse Transcription and Quantitative Polymerase Chain Reaction (RT-qPCR)

#### 4.5.1. RNA Isolation and Reverse Transcription

Total RNA from morula (5 morulas/group in 4 replicates) and hatched blastocysts (3 embryos/group in 4 replicates) was extracted using the PicoPure RNA Isolation kit (Life Technologies, Foster City, CA, USA) following the manufacturer’s protocol. Extracted RNA was stored at −80 °C until further analysis by qPCR. RNA concentration was quantified using a spectrophotometer (Nanodrop, Thermo Fisher Scientific, Waltham, MA, USA). cDNA synthesis was performed using a High-Capacity Reverse Transcription kit (Applied Biosystems, Foster City, CA, USA), following the manufacturer’s instructions. All samples were treated with DNase according to the manufacturer’s instructions before reverse transcription.

#### 4.5.2. Preamplification and qPCR

Gene expression analyses of bovine morulas and blastocysts were performed independently using Applied Biosystems^TM^ TaqMan^©^R Assays specific for *B. taurus* and based on Fontes et al. [38]. We analyzed the abundance of 52 transcripts using a panel of genes formatted to investigate embryonic competence and quality (apoptosis, oxidative stress, proliferation, and differentiation) and lipid metabolism in a microfluidic platform (Appendix A describing all the genes and their signaling pathways). Before qPCR thermal cycling, each sample was subjected to a sequence-specific preamplification process as follows: 1.25 mL assay mix (TaqMan^©^R Assay was pooled to a final concentration of 0.2× for each of the 52 assays), 2.5 mL TaqMan PreAmp Master Mix (Applied Biosystems, #4391128), and 1.25 mL cDNA (5 ng/mL). The reactions were activated at 95 °C for 10 min, followed by denaturation at 95 °C for 15 s, annealing, and amplification at 0 °C for 4 min for 14 cycles. These preamplified products were diluted fivefold (morula and blastocysts) prior to qPCR analysis. Assays and preamplified samples were transferred to an integrated fluidic circuit plate. For gene expression analysis, the sample solution preparation consisted of 2.25 mL cDNA (preamplified products), 2.5 mL of TaqMan Universal PCR Master Mix (2×, Applied Biosystems), and 0.25 mL of 20× GE Sample Loading Reagent (Fluidigm, South San Francisco, CA, USA); the assay solution included 2.5 mL 20× TaqMan Gene Expression Assay (Applied Biosystems) and 2.5 mL of 2× Assay Loading Reagent (Fluidigm). The 96.96 Dynamic Array^TM^ Integrated Fluidic Circuits (Fluidigm) chip was used for data collection. After priming, the chip was loaded with 5 mL each of the assay solutions and each sample solution and loaded into an automated controller that prepares the nanoliter-scale reactions. qPCR thermal cycling was performed in the Biomark HD System (Fluidigm), which involved one stage of Thermal Mix (50 °C for 2 min, 70 °C for 20 min, and 25 °C for 10 min) followed by a hot start stage (50 °C for 2 min and 95 °C for 10 min), 40 cycles of denaturation (95 °C for 15 s), primer annealing, and extension (both at 60 °C for 60 s).

### 4.6. Statistical Analysis

To estimate the fluorescence intensity (Experiment I), the results were evaluated regarding the data distribution, with these being non-parametric data; thus, the Kruskal–Wallis test was used, followed by Dunn’s post hoc test. The blastocyst rate was evaluated, and the hatching kinetics were tested for normal distribution. Data normality was assessed using the Shapiro–Wilk test. If the data had a normal distribution, Tukey’s test or a one-way ANOVA was applied. If they were non-parametric, data transformation (Log10) was applied, or the non-parametric Mann–Whitney test was applied. Data are presented as mean values and standard error of the mean (SEM) or median and 1st and 3rd interquartiles. In Experiment II, five replicates/group were performed, and Experiment III was performed with eight replicates/group (for blastocyst rate). All the above analyzes were performed with SigmaStat 4.0 software. Moderate statistical significance (i.e., an indicative of biological effect) was determined based on 0.01 < *p*-value ≤ 0.08, while strong significance was considered when *p*-value ≤ 0.01. Quantitative PCR data were assessed using the ∆Cq values relative to the geometric mean of the best reference genes among the 52-gene set, i.e., *GAPDH* and *ACTB*. Fold changes (FCs) were calculated using the 2^−∆Cq^ method [39]. All analyses were performed using SigmaStat 4.0 and MetaboAnalyst 6.0. The evaluation of the transcripts was initially performed with the univariate statistical analysis method using FCs and a *t*-test. In a second moment, we analyzed the data by multivariate methods, considering a non-supervised cluster analysis heatmap using the Euclidian/average linkage algorithm, Principal Component Analysis (PCA), and Partial Least Squares–Discriminant Analysis (PLS-DA) and their variations. Differences with probabilities less than *p* < 0.05, and/or FCs > 1.5, were considered significant.

## Figures and Tables

**Figure 1 ijms-25-10938-f001:**
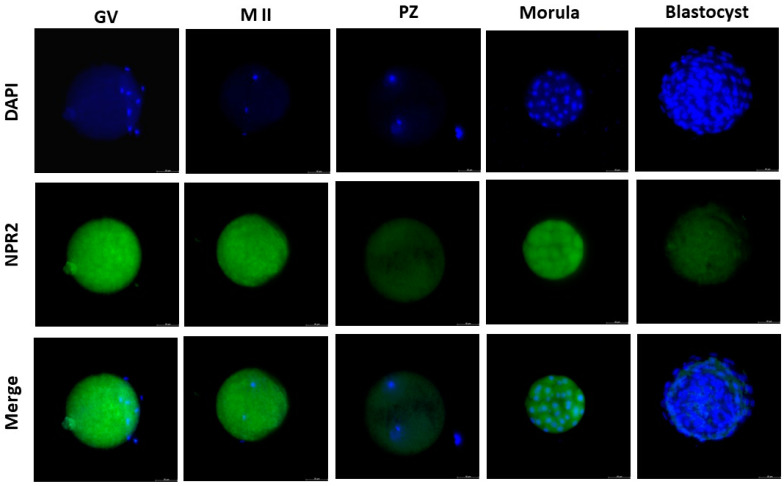
NPR2 localization in bovine oocytes and pre-implantation-stage embryos. The green color indicates NPR2 staining, and the blue color indicates nuclear staining (DAPI). NPR2 protein was expressed in bovine oocytes and embryos at all stages. GV, germinal vesicle; M II, metaphase II; PZs, presumptive zygotes; Morula, morula stage; Blastocyst, blastocyst stage. Bar = 50 µm.

**Figure 2 ijms-25-10938-f002:**
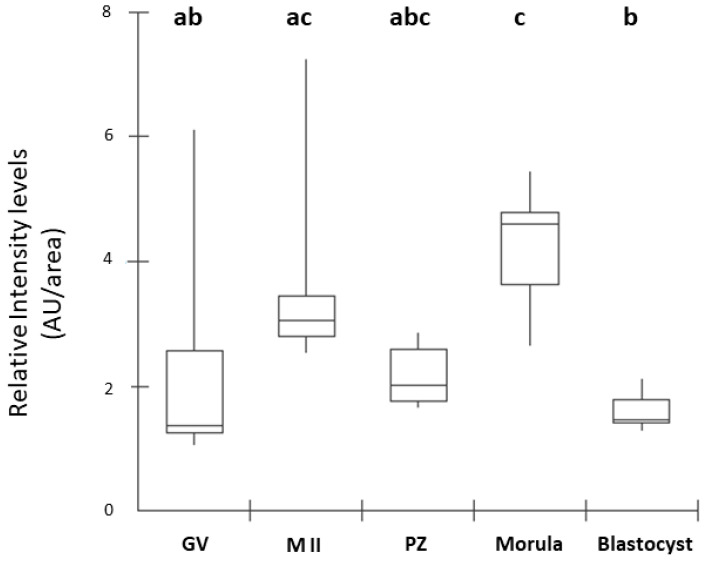
Box plot of fluorescence intensity of NPR2 in oocytes and pre-implantation-stage embryos. Results are presented as the median and 1° and 3° interquartile intervals of five replicates/stage using 8 structures in total. Different letters above each box represent significant differences (*p* ≤ 0.05). GV, germinal vesicle; M II, metaphase II; PZs, presumptive zygotes; Morula, morula stage; Blastocyst, blastocyst stage. AU, arbitrary units.

**Figure 3 ijms-25-10938-f003:**
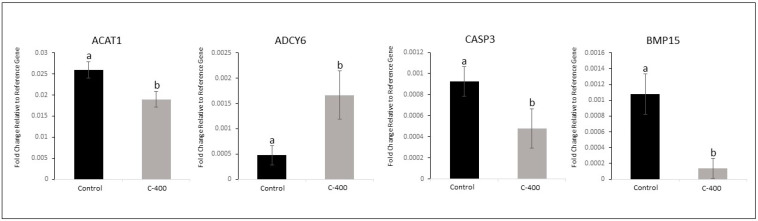
Effect of CNP treatment in IVC on differential gene expression in morula. Data represent the fold change in relative target abundance related to the reference gene. Downregulated transcription *ACAT1* (*p* = 0.040), *CASP3* (*p* = 0.082), and *BMP15* (*p* = 0.013) and upregulated transcription *ADCY6* (*p* = 0.057) with the addition of CNP (400 nM) on the D1 of the culture. Results are represented by least squares means ± SEM of four replicates/group. Different letters above each bar represent significant differences (*p* ≤ 0.08). Control (no treatment) and C-400 (400 nM of CNP).

**Figure 4 ijms-25-10938-f004:**
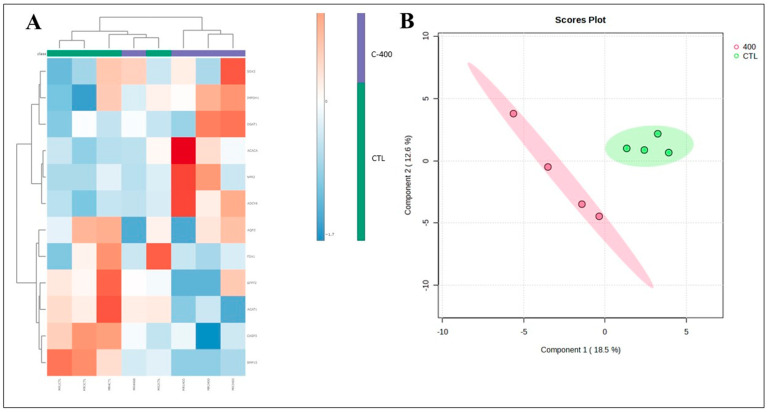
Multivariate analysis plots of the abundance of transcripts derived from untreated (control) and CNP-treated morula. (**A**) Cluster analysis heatmap showing transcriptional profiles abundance in only 12 genes most impacted from morula treated with 400 nM CNP and the control group. (**B**) Two-dimensional PLS-DA discrimination score plot between groups (5 morulas/group in 4 replicates).

**Figure 5 ijms-25-10938-f005:**
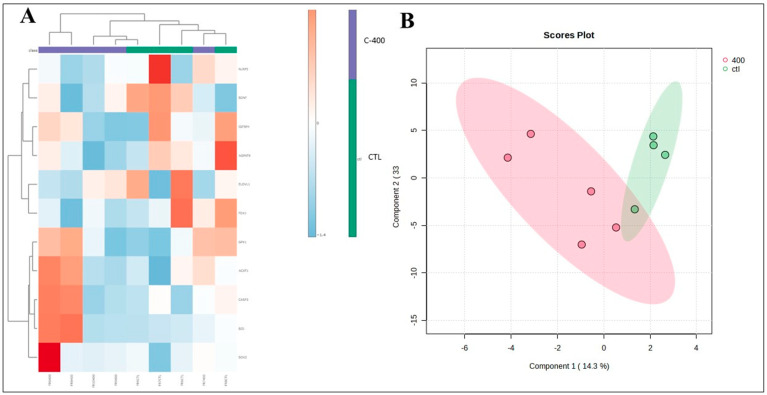
Multivariate analysis plots of the abundance of transcripts derived from untreated (control) and CNP-treated blastocyst. (**A**) Cluster analysis heatmap showing transcriptional profiles abundance in only 11 genes most impacted from blastocyst treated with 400 nM CNP and the control group. (**B**) Two-dimensional PLS-DA discrimination score plot between groups (3 embryos/group in 4 replicates).

**Figure 6 ijms-25-10938-f006:**
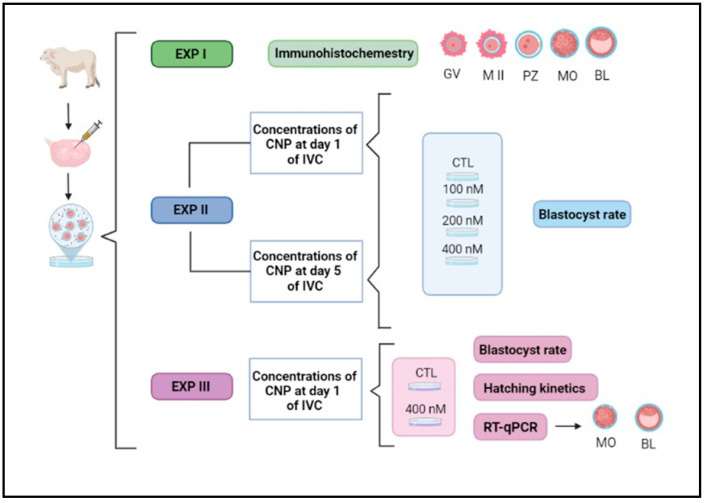
Illustrative experimental design. IVC, in vitro culture; EXP, experiment; GV, germinal vesicle; MII, metaphase II; PZs, presumptive zygotes; MO, morula; BL, blastocyst; CTL, control group; CNP, group treated with C-type natriuretic peptide. © 2024 BioRender.

**Table 1 ijms-25-10938-t001:** Rate of in vitro-produced bovine blastocyst supplemented with different concentrations of CNP from day 1 (D1) of IVC.

Group	*Cumulus*–Oocyte Complexes	PZs	Blastocyst
	*n*	*n*	*n* (% Mean ± SEM)
Control	91	90	29 (32.81 ± 14.24)
C-100	87	87	30 (34.14 ± 5.61)
C-200	94	93	33 (35.45 ± 5.06)
C-400	90	88	41 (46.09 ± 7.76)
*p*-value	-	-	0.082

C-100: 100 nM of CNP; C-200: 200 nM of CNP; C-400: 400 nM of CNP. PZs = presumptive zygotes. Data are the mean ± standard error of the mean (SEM) of 3 replicates.

**Table 2 ijms-25-10938-t002:** Rate of in vitro-produced bovine blastocyst supplemented with different concentrations of CNP from day 5 (D5) of IVC.

Group	*Cumulus*–Oocyte Complexes	PZs	Blastocyst
	*n*	*n*	*n* (% Mean ± SEM)
Control	182	180	60 (32.41 ± 5.45)
C-100	184	181	52 (28.62 ± 12.52)
C-200	186	183	48 (26.33 ± 5.64)
C-400	184	183	55 (29.17 ± 8.94)
*p*-value	-	-	0.743

C-100: 100 nM of CNP; C-200: 200 nM of CNP; C-400: 400 nM of CNP. PZs = presumptive zygotes. Data are the mean ± standard error of the mean (SEM) of 5 replicates.

**Table 3 ijms-25-10938-t003:** Evaluation of blastocyst rate (D7) and hatching kinetic (D7, D8, and D9) of embryos cultured with or without CNP (when added on day 1 of culture).

Group	PZs	Blastocyst Rate	Hatched Blastocyst D7	Hatched Blastocyst D8	Hatched Blastocyst D9	Total Hatched Blastocyst
	*n*	*n* (% Mean ± SEM)	*n* (% Median (1st, 3rd)) *	*n* (% Mean ± SEM)	*n* (% Mean ± SEM)	*n* (% Mean ± SEM)
Control	979	332 (34.13 ± 2.11)	4 [0.98 (0.00, 2.11)]	84 (25.76 ± 4.96)	62 (17.88 ± 2.75)	150 (44.80 ± 5.51)
C-400	1026	331 (32.55 ± 1.14)	3 [0.00 (0.00, 1.06)]	71 (20.58 ± 3.79)	68 (20.00 ± 2.09)	142 (41.34 ± 5.34)
*p*-value	-	0.52	0.57	0.42	0.55	0.66

* Data are presented as median and 1st and 3rd interquartiles. CNP-400: 400 nM de CNP. PZs = presumptive zygotes. Data are the mean ± standard error of the mean (SEM) of 8 replicates, except for *.

**Table 4 ijms-25-10938-t004:** Upregulated and downregulated transcription observed in morula stage after treatment with CNP. The relative abundance of transcripts was selected based on the fold change analysis (with a magnitude greater than 1.5 time, that is, with a threshold >1.5). The values shown were calculated as the ratio of the control group to the treated group.

	Gene Symbol	Definition	Fold Change
	IMPDH1	GTP/cGMP	0.663
	CD40	Apoptosis	0.627
Upregulated	ADCY6	cAMP/meiotic arrest	0.303
	ELF5	Cell differentiation/trophectoderm	0.199
	NPR2	cGMP/meiotic arrest	0.190
	BMP15	Oocyte maturation/follicular development	7.008
	FSHR	Follicle stimulating hormone receptor/gonad development	3.246
	NRP2	Cell survival/follicular development	1.908
Downregulated	NANOG	Pluripotency (ICM/TE)/when overexpressed, promotes cells to enter the S phase and proliferation	1.878
	GFPT2	Oxidative stress	1.866
	CASP3	Apoptosis	1.835
	HSPA1A	Cell survival/facilitates DNA repair	1.579

**Table 5 ijms-25-10938-t005:** Upregulated and downregulated transcription observed in blastocyst stage after treatment with CNP. The relative abundance of transcripts was selected based on the fold change analysis (with a magnitude greater than 1.5 times, that is, with a threshold >1.5). The values shown were calculated as the ratio of the control group to the treated group.

	Gene Symbol	Definition	Fold Change
	*HSPA5*	Folding and assembly of proteins in the endoplasmic reticulum/degradation of misfolded proteins	0.637
Upregulated	*SOX2*	Pluripotency/chromatin binding/DNA methylation	0.571
	*CASP3*	Apoptosis	0.513
	*BID*	Apoptosis/pro-apoptotic	0.408
	*BDNF*	Supporting meiotic progression	1.966
	*NLRP5*	Maternal oocyte protein/required for normal early embryogenesis	1.868
	*AGPAT9*	Predict embryo quality/lipid metabolism	1.794
Downregulated	*IGFBP4*	Either inhibit or stimulate the growth promoting effects of the IGFs	1.700
	*ELOVL4*	Fatty acid biosynthesis, elongation, endoplasmic reticulum	1.725
	*ELOVL1*	Fatty acid biosynthesis, elongation, endoplasmic reticulum	1.670
	*FDX1*	Synthesis steroid hormones/catalyzes cholesterol cleavage	1.533

## Data Availability

The raw data supporting the conclusions of this article will be made available by the authors, without undue reservation, if requested.

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
