# Peer review of "Developmental and Molecular Effects of C-Type Natriuretic Peptide Supplementation in In Vitro Culture of Bovine Embryos"

_ijms, 2024, doi:10.3390/ijms252010938_

Round 1
Reviewer 1 Report
Comments and Suggestions for Authors
This study focused on exploring the role of C-type natriuretic peptide (CNP) in bovine preimplantation embryo development, coupled with demonstrating the effect of CNP on embryo competence, including transcripts related to embryonic metabolism in morula and blastocyst stage. This study is well designed and thoroughly investigating the effect of different CNP concentration in bovine embryo quality, and demonstrated that high level of CNP has positive effect on blastocyst rate, which augments the quality of this study.
However, the quality of the manuscript could be improved further with attention to several issues:
One of the issues in this study is that author mentioned CNP receptor (natriuretic peptide receptor, NPR2) and detected its expression in different stages, but it is not explored whether the effects of CNP is through NPR2 in bovine embryos. As NPR2 level peaks during morula stage, while the effects of CNP is highlighted during blastocyst. A knockdown or knockout experiment is necessary to verify this hypothesis.
Also, it is shown that CNP treatment increased the blastocyst rate, but no further characterization was performed. I am curious about the cell number and lineage composition in CNP treated blastocyst and hatched blastocyst. Additionally, if 400nM CNP was not detrimental for embryos, what about higher level? Considering lower level (~200nM) did not affect the developmental rate, a higher concentration might have better effects.
Reviewer 2 Report
Comments and Suggestions for Authors
The manuscript is captivating and meticulously prepared. The authors delineate Developmental and molecular effects of CNP supplementation in in vitro culture of bovine embryos. The paper is exquisitely written, and I have no apprehensions regarding the language. I acknowledge the considerable effort expended in these studies, and I compliment the authors for their endeavors. Nevertheless, as a reviewer, I am obliged to pose some questions that I deem critical to the overall assessment of the research.
Comments 1 It is recommended that authors be able to add a line number to each line.
Comments 2 The introduction section can expand the narrative, such as a brief introduction to the CNP and its reported role. In addition, the meaning and purpose of this experiment can be added to the last sentence of the introduction.
Comments 3 Can it be supplemented In 2.1 In vitro production of embryos how the authors separated COCs from granulosa cells.
Comments 4 Why 5 days for CNP treatment? Is there any basis for the choice of CNP concentration? What is the concentration of endogenous CNP in general?
Comments 5 In the Results section, Experiment I Experiment â…¡ Experiment â…¢ is generally not used as the name of the subheading.
Round 2
Reviewer 1 Report
Comments and Suggestions for Authors
Authors addressed my comments, and the paper is fine to be published in current form.